# An in depth look at the Procrustes-Wasserstein distance: properties and barycenters

**Davide Adamo**[1,2]   **Marco Corneli**[1,2]   **Manon Vuillien**[1]   **Emmanuelle Vila**[3]

## Abstract

Due to its invariance to rigid transformations such as rotations and reflections, Procrustes-Wasserstein (PW) was introduced in the literature as an optimal transport (OT) distance, alternative to Wasserstein and more suited to tasks such as the alignment and comparison of point clouds. Having that application in mind, we carefully build a space of discrete probability measures and show that over that space PW actually *is* a distance. Algorithms to solve the PW problems already exist, however we extend the PW framework by discussing and testing several initialization strategies. We then introduce the notion of PW barycenter and detail an algorithm to estimate it from the data. The result is a new method to compute representative shapes from a collection of point clouds. We benchmark our method against existing OT approaches, demonstrating superior performance in scenarios requiring precise alignment and shape preservation. We finally show the usefulness of the PW barycenters in an archaeological context. Our results highlight the potential of PW in boosting 2D and 3D point cloud analysis for machine learning and computational geometry applications.

## 1. Introduction

In force of its capability to find correspondences between sets of objects, in the last decade computational optimal transport (OT, Peyré et al., 2019) has become more and more ubiquitous in machine learning. Notable examples relate to learning tasks from (almost) any data type including images (Solomon et al., 2015; Feydy et al., 2017),

[1]Université Côte d'Azur, UMR 7264 CEPAM, CNRS, Nice, France [2]Université Côte d'Azur, Inria, CNRS, Laboratoire J.A. Dieudonné, Maasai team, Nice, France [3]Université Lumiére Lyon II, UMR 5133 Archéorient CNRS, Lyon, France. Correspondence to: Davide Adamo <davide.adamo@univ-cotedazur.fr>.

*Proceedings of the 42nd International Conference on Machine Learning*, Vancouver, Canada. PMLR 267, 2025. Copyright 2025 by the author(s).

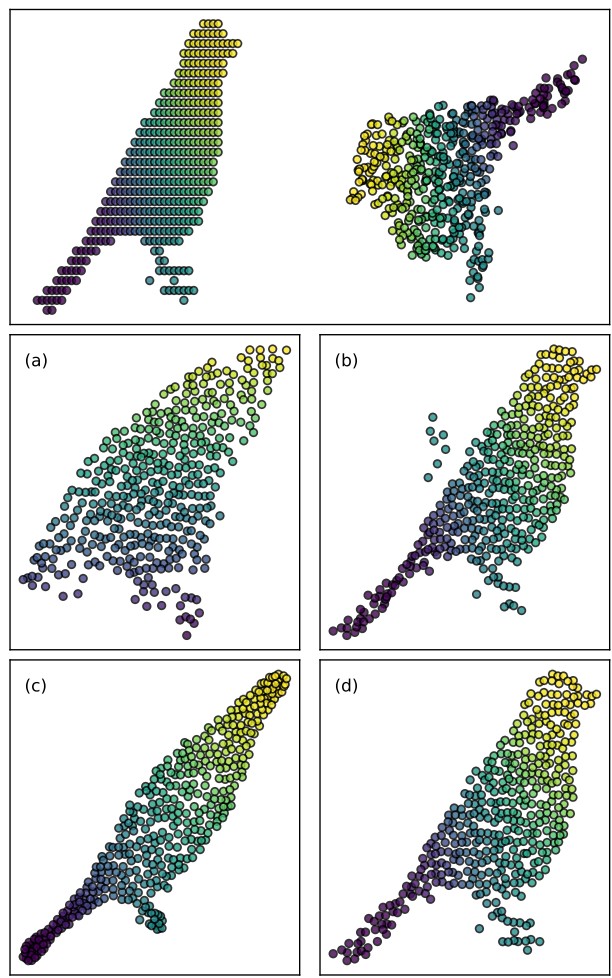

*Figure 1.* (Top) Two point clouds representing a bird shape in different position. OT barycenters using (a) Exact Free Wasserstein (Cuturi & Doucet, 2014) (b) Gromov-Wasserstein (Peyré et al., 2016) with MDS (Borg & Groenen, 2007) (c) Gromov-Wasserstein with TSNE (Van der Maaten & Hinton, 2008) (d) Procrustes-Wasserstein (our).

graphs (Vayer et al., 2019a; Vincent-Cuaz et al., 2021), shapes (Eisenberger et al., 2020) or text (Zhang et al., 2017; Grave et al., 2019). More generally and possibly more importantly OT allows one to assess the distance between

probability distributions thus leading to applications in machine learning that go far beyond the comparison of sets of objects, such as domain adaptation (Courty et al., 2016) or adversarial training (Arjovsky et al., 2017), just to cite some. However, here we keep the focus on the first framework we cited (i.e. data alignment and matching) since the applications we discuss in this work are of that kind.

Based on the modern formulation of Kantorovich (1942), the standard optimal transport tool to compare two sets of objects is the Wasserstein distance. If we assume that each set is a point cloud, in order to fix the ideas, adopting the Wasserstein distance to quantify the similarity between the clouds specifically requires to compute the Euclidean distance (or other) between the points of the first cloud *and* those of the second. Since, moreover, each point is equipped with a probability mass defining its importance *within* its cloud, we can say that the Wasserstein distance takes into account both the geometry and the distributional properties of data. However, the Wasserstein distance suffers from some limitations that makes it unfit to some applications such as point cloud matching. Indeed, it is sensitive to the way the two clouds are embedded in the space and in particular to isometries. To address some of the limitations of the Wasserstein distance, Mémoli (2011) introduced the Gromov-Wasserstein (GW) distance. Unlike standard OT frameworks, which assume that the compared probability measures are supported on a shared metric space, GW compares distributions defined on distinct spaces. In the point cloud matching examples, GW requires computing two pairwise distance (or similarity) matrices between the points *within* each cloud. Points not being in the same cloud are never compared explicitly. GW is invariant to isometries and particularly suitable for the comparison of data sets with unknown correspondences or in different coodinate systems. However, GW has (at least) two main drawbacks: i) its rather prohibitive computational cost, although some solutions exist (Vayer et al., 2019b; Chowdhury et al., 2021) and ii) the GW barycenters are *still* pairwise distance/similarity matrices. If one wishes to represent them in the original features domain, dimensionality reduction techniques are needed. This last drawback can be severe when computing mean shapes where a high fidelity to the original is required (see Figure 1).

Mixing Procrustes and Wasserstein costs was recently done (Zhang et al., 2017; Grave et al., 2019) in order to introduce into the Wasserstein optimization problem invariances to global transformations such as rotations and reflections in the space. In this sense Procrustes-Wasserstein (PW) can be seen as a compromise between GW (with whom it shares some invariances) and Wasserstein (since the two measures are directly compared with each other).

**Related works.** Among the earliest PW formulations, Zhang et al. (2017); Grave et al. (2019) aimed at jointly estimating an orthogonal and a permutation matrix to align word embeddings across different languages. Differently from Zhang et al. (2017), which initialized the orthogonal matrix using an adversarial training phase, Grave et al. (2019) proposed a convex relaxation of the initialization by reformulating the problem over the convex hull. Alvarez-Melis et al. (2019) extended the previous works by incorporating global invariances directly into the optimization process. Their approach is not limited to invariances with respect to isometries but generalizes to broader invariance classes (characterized by Schatten $p$-norm ball) up to the recovery of Gromov-Wasserstein. This extension is particularly useful in scenarios where data are not simply related by rigid transformations. Additionally, employing a convexity-annealing strategy and considering a relaxed PW version, they eliminate the need for an ad-hoc initialization, avoiding strong dependence on an initial guess. In contrast with previous works, two-sided PW (TWP, Jin et al., 2021) adopts a two-fold transformation on both the source and target measures. Such an extension enables to handle data that lie in distinct spaces, transporting them into a common latent space. The optimal solution is obtained by solving a component-wise convex optimization problem, combining two-sided Procrustes Analysis with a relaxed Wasserstein formulation. Aboagye et al. (2022) tackle the computational limit of PW by proposing a quantized version of the problem (qWP). The quantization step that discretizes the distributions enables for the joint estimate of the alignment and transformation. qWP leverages a quantization procedure inspired by Grave et al. (2019), such as $k$-means++, and reduces the problem to linear programming (LP). This technique not only simplifies the computation but also enhances the approximation quality of OT solvers, thus leading to a more efficient solutions with a fixed computational cost. Finally, Even et al. (2024) approach the problem of matching pairs of distributions using PW distances from a theoretical perspective, providing convergence guarantees for the ML estimators of both the transport plan and the isometry. In more detail, they restrict their focus on discrete distributions with the same number of points in the support, further assuming that one distribution can be obtained from the other through a permutation and isometry of the support and the addition of Gaussian noise. The corresponding OT problem falls under the Monge formulation and instead of looking for doubly stochastic plans, they look for permutation matrices.

**Contribution of our work.** Despite the heterogeneous use of the PW cost in the above mentioned works, to the best of our knowledge i) it was never showed that Procrustes-Wasserstein distance actually *is* a distance; ii) PW barycenters were never defined/learned from the data. With a focus on scenarios where the objects to compare are geometric

shapes represented as point clouds (and hence working with discrete measures) the main contribution of this paper is twofold: we define a quotient space of discrete measures over which PW is a distance and we provide an estimation algorithm for the PW barycenter. We then show that one of the main advantages of PW is its capability to produce very faithful barycenters in particular conditions. In the illustrative example in Figure 1, two birds differ in both number of vertices and pose (rotation and/or reflection). As it can be seen, among the three tested OT methods, the PW barycenter result in more consistent geometric characteristics. We finally present a concrete application of the PW barycenters to detect morphological changes on archaeozoological data.

The paper is organized as follows: we provide a background on the PW problem and the formal definitions and proof where PW is a distance in section 2. We then introduce the PW barycenters and the algorithm to compute it in section 3. We investigate different intializations for specific match in point clouds and a clustering application on our barycenter in section 4. We conclude presenting a concrete real-world application in section 5.

## 2. Procrustes-Wasserstein: an OT distance

**Notation.** We denote by $\Sigma_n$ the $n-1$ probability simplex. So when saying that $\mathbf{p} := (p_1, \ldots, p_n) \in \Sigma_n$, we mean $p_i \geq 0$ for all $i$ and $\sum_{i=1}^n p_i = 1$. We denote by $\langle \cdot, \cdot \rangle_F$ the Frobenious dot product, hence $\langle A, B \rangle_F := \text{trace}(B^T A)$, with $A, B$ two compatible matrices. The set of the orthogonal matrices of order $d$ is denoted by $\mathcal{O}(d)$.

Consider two matrices $X \in \mathbb{R}^{n \times d}$ and $Y \in \mathbb{R}^{m \times d}$, where $\mathbf{x}_i$ (respectively $\mathbf{x}^j$) is the i-th row (j-th column) of $X$. Similarly for $Y$. We attach two discrete probability measures $\mu_X$ and $\mu_Y$ to $X$ and $Y$, respectively:

$$\mu_X = \sum_{i=1}^n p_i \delta_{\mathbf{x}_i}, \qquad \mathbf{p} \in \Sigma_n \quad (1)$$

and

$$\mu_Y = \sum_{j=1}^m q_j \delta_{\mathbf{y}_j}, \qquad \mathbf{q} \in \Sigma_m.$$

Given an orthogonal matrix $P \in \mathcal{O}(d)$, we denote by $\mu_{YP}$ the measure defined on the transformed support of $Y$, namely $\mu_{YP} := \sum_{j=1}^m q_j \delta_{\mathbf{y}_j P}$.

Given $W_2(\mu_X, \mu_Y)$, the 2-Wasserstein distance between $\mu_X$ and $\mu_Y$, we attack the following minimization problem

$$\min_{P \in \mathcal{O}(d)} W_2^2(\mu_X, \mu_{YP}) = \min_{\substack{P \in \mathcal{O}(d) \\ \Gamma \in \Pi(\mathbf{p}, \mathbf{q})}} \langle C_P(X, Y), \Gamma \rangle_F \quad (2)$$

where $\Pi(\mathbf{p}, \mathbf{q})$ is the set of the admissible transport plans, i.e.

$$\Pi(\mathbf{p}, \mathbf{q}) = \{\Gamma \in \mathbb{R}_+^{n \times m} | \Gamma \mathbf{1}_m = \mathbf{p}, \Gamma^T \mathbf{1}_n = \mathbf{q}\}$$

and $C_P(X, Y) \in \mathbb{R}_+^{n \times m}$ with

$$(C_P(X, Y))_{ij} = \|\mathbf{x}_i - \mathbf{y}_j P\|_2^2.$$

The above minimization problem is a generalization of the one described in Grave et al. (2019) and can be seen as a particular case of the one discussed in Alvarez-Melis et al. (2019).

By definition of $C_P(X, Y)$ it is easy to show that

$$C_P(X, Y) = R_X + R_Y - 2XP^T Y^T,$$

where the i-th row of $R_X \in \mathbb{R}^{n \times m}$ is $(\|\mathbf{x}_i\|_2^2, \ldots, \|\mathbf{x}_i\|_2^2)$ and the j-th column of $R_Y \in \mathbb{R}^{n \times m}$ is $(\|\mathbf{y}_j\|_2^2, \ldots, \|\mathbf{y}_j\|_2^2)^T$. By plugging this into Eq. (2) and thanks to the bilinearity of $\langle \cdot, \cdot \rangle_F$ we get

$$\langle C_P(X, Y), \Gamma \rangle_F = \langle R_X + R_Y, \Gamma \rangle_F - 2\langle XP^T Y^T, \Gamma \rangle_F$$
$$= \langle \mathbf{u}, \mathbf{p} \rangle + \langle \mathbf{v}, \mathbf{q} \rangle - 2\langle XP^T Y^T, \Gamma \rangle_F, \quad (3)$$

where $\mathbf{u} \in \mathbb{R}^n$ is such that $u_i = \|x_i\|_2^2$ and $\mathbf{v} \in \mathbb{R}^m$ such that $v_j = \|y_j\|_2^2$. As such, the minimisation problem in Eq. (2) is equivalent to

$$\max_{\substack{P \in \mathcal{O}(d) \\ \Gamma \in \Pi(\mathbf{p}, \mathbf{q})}} \langle XP^T Y^T, \Gamma \rangle_F. \quad (4)$$

We now consider the set $\mathcal{M}_d$ of all discrete measures of the same form as in Eq. (1). Namely, the generic $\mu_X \in \mathcal{M}_d$ is a measure supported on some $X \in \mathbb{R}^{n \times d}$, for some finite $n$ and a *fixed* $d$ and for some probability vector $\mathbf{p}$. With a slight abuse of notation, given a permutation $\sigma$ in $\mathcal{S}^{(n)}$, the set of all possible permutations of $n$ elements, we denote by $\sigma(X) = (\mathbf{x}_{\sigma(1)}^T, \ldots, \mathbf{x}_{\sigma(n)}^T)^T$ the matrix $X$ after the permutation of its rows according to $\sigma$. Similarly, we denote by $\sigma(\mathbf{p}) = (p_{\sigma(1)}, \ldots, p_{\sigma(n)})$ the permuted histogram. We introduce the following equivalence relation on $\mathcal{M}_d$

$$\mu_{X_1} \sim \mu_{X_2} \quad \text{if} \quad \exists P \in \mathcal{O}(d), \exists \sigma \in S^{(n)}$$

such that $X_1 = \sigma(X_2)P$ and $\mathbf{p}_1 = \sigma(\mathbf{p}_2)$.

Thus, $\mu_{X_1} \sim \mu_{X_2}$ if and only if they share the same probability vector, up to a permutation, and the same support up to the same permutation of the points and a rigid transformation (rotation, reflection or a combination of both).

If we denote

$$PW_2(\mu_X, \mu_Y) := \left( \min_{\substack{P \in \mathcal{O}(d) \\ \Gamma \in \Pi(\mathbf{p}, \mathbf{q})}} \langle C_P(X, Y), \Gamma \rangle_F \right)^{1/2}, \quad (5)$$

then

**Algorithm 1** PW problem

1: **Input:** Locations and histograms $(X, \mathbf{p})$, $(Y, \mathbf{q})$; initial correspondences $\Gamma_0$.
2: *%% Initialization*
3: $U\Sigma V^T \leftarrow \text{SVD}(Y^T \Gamma_0^T X)$
4: $P_0 \leftarrow UV^T$, $P \leftarrow P_0$
5: **while** not converged **do**
6:    $C_P \leftarrow \text{cost}(X, YP)$
7:    *%% Update matching*
8:    $\Gamma \leftarrow \text{EMD}(C, \mathbf{p}, \mathbf{q})$ *%% Earth Mover Distance*
9:    *%% Update P*
10:    $U\Sigma V^T \leftarrow \text{SVD}(Y^T \Gamma^T X)$
11:    $P \leftarrow UV^T$
12: **end while**
13: **Return:** $\Gamma^*$, $P^*$

**Theorem 2.1.** *$PW_2(\cdot, \cdot)$ is a distance on $\mathcal{M}_d/\sim$.*

The proof of the above theorem is in Supplementary Material A. Moreover we have the following

**Corollary 2.2.** *For all $\mu_X, \mu_Y$ in $\mathcal{M}_d$ it holds that $PW_2(\mu_X, \mu_Y) \leq W_2(\mu_X, \mu_Y)$.*

*Proof.* It suffices to note that

$$
\begin{aligned}
PW_2(\mu_X, \mu_Y) := &\min_{P \in \mathcal{O}(d)} W_2(\mu_X, \mu_{YP}) \\
&\leq W_2(\mu_X, \mu_{YI_d}) = W_2(\mu_X, \mu_Y).
\end{aligned}
$$

$\square$

## 3. Procrustes-Wasserstein barycenter(s)

Now that we established that $PW_2$ is a distance on $\mathcal{M}_d/\sim$, consider $r$ empirical measures measures $\mu_{X_1}, \ldots, \mu_{X_r}$, in $\mathcal{M}_d$, with supports $\{X_j\}_{j=1}^r$ and probability vectors $\{\mathbf{p}_j\}_{j=1}^r$. We look for a barycenter $\mu_X$ with unknown support $X \in \mathbb{R}^{n \times d}$ and weights $\mathbf{p}$ given by the solution to the following problem

$$
f(\mathbf{p}, X) := \frac{1}{r} \sum_{j=1}^r PW_2^2(\mu_X, \mu_{X_j}). \tag{6}
$$

In a general setting, we might consider positive weights $\lambda_j$ associated with each measure $\mu_{X_j}$, with $\boldsymbol{\lambda} := (\lambda_1, \ldots, \lambda_r) \in \Sigma_r$. For simplicity, we present the case $\lambda_j = \frac{1}{r}$.

### 3.1. Differentiability of $f(\mathbf{p}, X)$ with respect to $X$

In this section we assume that $\mathbf{p}$ is known. Let $X \in \mathbb{R}^{n \times d}$ and $Y \in \mathbb{R}^{m \times d}$. Consider the transport cost as a function

of $X$ as outlined in Equation (3). The minimization of $PW_2^2(\mu_X, \mu_Y)$ with respect to $X$ can be developed as

$$
\begin{aligned}
\min_X PW_2^2(\mu_X, \mu_Y) &= \min_X \min_{P, \Gamma} \langle C_P(X, Y), \Gamma \rangle_F \\
&= \min_X \left( \langle \mathbf{u}, \mathbf{p} \rangle + 2 \min_{P, \Gamma} \langle -X, \Gamma Y P \rangle_F \right),
\end{aligned} \tag{7}
$$

where constant terms in $Y$ and $\mathbf{q}$ are discarded. While the first term is a convex quadratic function of $X$ (since $u_i = \|x_i\|_2^2$), the second term renders the optimisation of $PW_2^2(\mu_X, \mu_Y)$ with respect to $X$ non-convex. Thus, the best we can do is to look for local minima via Newton-Raphson. Denote by $(P^*, \Gamma^*)$ the optimal alignment and transport plan for $PW_2^2(\mu_X, \mu_Y)$. Calling $g(X)$ the objective function in Eq. (7)

$$
g(X) := \langle \mathbf{u}, \mathbf{p} \rangle - 2 \langle X, \Gamma^* Y P^* \rangle_F,
$$

the gradient and the Hessian of $g(\cdot)$ with respect to $X$ are

$$
\nabla_X g = 2\text{diag}(\mathbf{p})X - 2\Gamma^* Y P^*,
$$

and

$$
H_X g = 2\text{diag}(\mathbf{p}).
$$

Thus, the update of $X$ reads

$$
\begin{aligned}
X^{(k+1)} &= X^{(k)} - \underbrace{(H_X g(X^{(k)}))^{-1} \cdot \nabla_X g(X^{(k)})}_{\text{Newton step}} \\
&= X^{(k)} - (X^{(k)} - \text{diag}(\mathbf{p}^{-1}) \Gamma^* Y P^*) \\
&= \text{diag}(\mathbf{p}^{-1}) \Gamma^* Y P^*.
\end{aligned} \tag{8}
$$

The update formula provides a meaningful geometric interpretation. The matrix $\text{diag}(\mathbf{p}^{-1})\Gamma^*$, whose $n$ rows belong to the simplex $\Sigma_m$, computes weighted barycenters of points in $Y$, with weights defined by the optimal transport plan. This is analogous to the Wasserstein barycenter update (Cuturi & Doucet, 2014), where each point in $Y$ contributes to the updated locations in $X$ proportionally to $\Gamma^*$. However, in the PW framework, the additional right multiplication by $P^*$ allows for a simultaneous optimal alignment of the barycenter.

The steps to optimize $f(\mathbf{p}, X)$ with respect to the locations $X$ are outlined in Algorithm 2. Solving Problem (6) involves computing $r$ *independent* PW distances between the barycenter ($\mu_X$) and the measures $\mu_{X_j}$. Thus, the first step (lines 4-5) consists into solving all $PW_2^2(\mu_X, \mu_{X_j})$ and finding $r$ solutions $(\Gamma_j^*, P_j^*)$ following the iterative scheme introduced in (Grave et al., 2019) that we report here in Algorithm 1 for completeness. The second step (line 7) updates the locations of the barycenter using the update formula in Eq. (8).

**Algorithm 2** Procrustes-Wasserstein barycenter (PWB)

1: **Input:** Locations $X_j \in \mathbb{R}^{n_j \times d}$ and histograms $\mathbf{p}_j \in \mathbb{R}^{n_j}$ for $j = 1, \ldots, r$; initial barycenter locations $X_0$; barycenter histogram $\mathbf{p}$
2: $X = X_0$
3: **while** not converged **do**
4:    **for** $j \in (1, \ldots, r)$ **do**
5:      $(\Gamma_j^*, P_j^*) \leftarrow PW_2(X, \mathbf{p}; X_i, \mathbf{a}_j)$
6:    **end for**
7:    $X = X + \frac{1}{r}\left(\sum_{i=1}^{r} \Gamma_i^* X_i P_i^*\right) \cdot \mathrm{diag}(\mathbf{p}^{-1})$
8: **end while**
9: **Return:** $X^*$

### 3.2. Differentiability of $f(\mathbf{p}, X)$ with respect to $\mathbf{p}$

Despite the obvious difference between the minimisation problem in Eq. (6) and its Wasserstein counterpart illustrated in Cuturi & Doucet (2014), it can be observed that

$$f(\mathbf{p}, X) = \frac{1}{r} \sum_{j=1}^{r} PW_2^2(\mu_X, \mu_{X_j})$$

$$= \frac{1}{r} \sum_{j=1}^{r} W_2^2(\mu_X, \mu_{X_j P_j^*}).$$

where, $P_j^*$ is the optimal isometry aligning $\mu_{X_j}$ with the barycenter. Denoting $\hat{X}_j := X_j P_j^*$, the analogy with the Wasserstein dual LP formulation is straightforward

$$\max_{\alpha_j, \beta_j} \langle \alpha_j, \mathbf{p}\rangle + \langle \beta_j, \mathbf{p_j}\rangle, \qquad (9)$$

where in the PW framework the couplings $(\alpha_j, \beta_j)$ must satisfy

$$\alpha_{j,i} + \beta_{j,k} \leq (C_{P_j^*})_{ik} = \|x_i - \hat{x}_{j,k}\|^2,$$

where $(C_{P_j^*})$ is the cost matrix incorporating the orthogonal alignment and $\hat{x}_{j,k}$ denotes here the $k$-th row of $\hat{X}_j$. Eq. (9) is a linear programming (LP) problem for each $j$, with constraints defined by $(C_{P_j^*})$. The optimization of $f(\mathbf{p}, X)$ with respect to $\mathbf{p}$ can be approached analogously to Cuturi & Doucet (2014), leveraging the solutions of the dual problems, e.g. $\boldsymbol{\alpha} := \frac{1}{r}\sum_{j=1}^{r} \alpha_j^*$. For completeness, we provide Algorithm 3 in the supplementary material, detailing the procedure for the optimization with respect to $\mathbf{p}$.

When pursuing the joint optimization of Eq. (6) with respect to $(\mathbf{p}, X)$, the outlined strategy remains the one presented in Algorithm 2, except for an additional equation after line 7 updating the weights $\mathbf{p}$ according to Algorithm 3.

## 4. Experiments

All the point clouds considered in this section are assumed to be centered at they Euclidean barycenter and scaled in

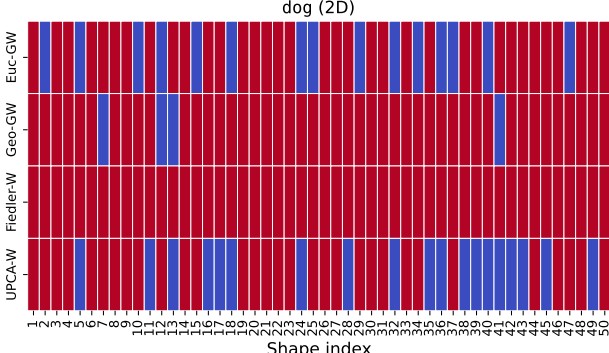

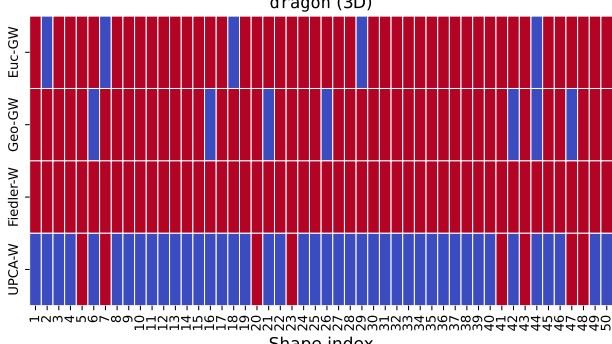

*Figure 2.* Convergence comparison between different initialization approaches across randomly generated shapes. Each row corresponds to a different initialization method while each column corresponds to a run of the Algorithm 1 with a different shape. Red cells indicates successful convergence of the matching (in terms of rotation/reflection and couplings), while blue cells denotes failure.

such a way to be enclosed the 1D or 2D unit ball. We leave for future works extensions of the PW framework accounting for translations and scaling. The OT solvers used in the implementations are based on the POT toolbox (Flamary et al., 2021). The code is available at https://github.com/DavideAdamo98/PW-bary.

### 4.1. Initialization for point cloud matching

It is well known that a primary challenge in the computation of PW lies in its initialization (Grave et al., 2019; Alvarez-Melis et al., 2019). In this section, we inspect several initialization strategies of $\Gamma_0$ (Algorithm 1) for PW in the context of 2D/3D point cloud matching.

Let us consider a pivot measure $\mu_{X_1}$, either representing a 2 or 3-dimensional point cloud. Since it is assumed that each point is equipped with the same (uniform) probability mass, with a sligth abuse of notation we identify $\mu_{X_1}$ with $X_1$. We generate 50 clouds by randomly adding extra vertices, Gaussian noise and vertex permutation to $X_1$. We also

include a random rotation and reflection. We thus generate $X_2^i$ for $i = 1, \ldots, 50$ that underline the same geomertric structure of $X_1$ (e.g. they represent a perturbed versions of the pivot). We look for a pairwise clouds registration, in terms of global alignment and couplings. We test different approaches, with the objective to compute $\Gamma_0$.

1. **Euc-GW.** Gromov-Wasserstein based on Euclidean pairwise distances is computed for each pair of point clouds and $\Gamma_0$ is set equal to the optimal GW plan.

2. **Geo-GW.** Same as before but with geodesic pairwise distance in place of the Euclidean.

3. **Fiedler-W.** Fiedler vector (Fiedler, 1973) is the eigen-vector associated with the algebraic connectivity (i.e. the second-smallest eigenvalue) of the Laplacian matrix of a connected graph. Since point clouds can be easily transformed into graphs (Cover & Hart, 1967; Preparata & Shamos, 2012) $\mathcal{G}$. We propose to resort to a Wasserstein matching between Fiedler vectors to initialise $\Gamma_0$. More specifically, for a fixed $i$, we compute the Fiedler vectors of $X_1$ and $X_2^i$, denoted as $f_1$ and $f_2^i$, respectively, and we standardize them. Furthermore, we compute both the Wasserstein distance between $(f_1, f_2^i)$ and $(f_1, -f_2^i)$ (to account for the vectors orientation). The transport plan yielding the smaller distance determines $\Gamma_0$.

4. **UPCA-W.** Given two point clouds, $X$ and $Y$, the first step involves computing the eigenvector matrices $Q_X$ and $Q_Y$, of their covariance matrices. The multiplica-tion $XQ_X$ (resp. $YQ_Y$) leads to a matrix $X'$ (respec-tively $Y'$) that is uncorrelated, e.g. the principal axes of $X'$ and $Y'$ correspond, up to the directions, to the stan-dard coordinate axes of the $d$-dimensional Euclidean space. Moreover, fixing $X$, the matrix $YQ_X^T Q_Y$ brings $Y$ into the same (principal component) basis as $X$, once more up to the direction of the axes. At this stage, a Wasserstein matching can be performed between $X$ and $YQ_X^T Q_Y$. In the case of $d = 2$, there are $2^2$ possi-ble combinations of directions to check, requiring the resolution of four independent Wasserstein problems. Similarly, for $d = 3$, we must solve $2^3$ Wasserstein problems. As with Fiedler-W, the transport plan asso-ciated to the smallest distance defines the initialization $\Gamma_0$.

Convergence results of the Algorithm 1 for the four pre-sented initialization techniques are summarized in Figure 2. Red colour for the cells denotes convergence to the global minimum (matching succeeded) while blue colour denotes failure (convergence to local minima). We observe that GW initializations generally lead to a good success rate.

However, despite their invariance to isometries, there are instances where the GW transport plan fails to establish the correct couplings. In cases where the data underline specific geometric structure GW could reveal optimal, however its computational cost makes it use clearly prohibitive when working with larger point clouds. In contrast, the Fiedler-W initialization consistently ensures robust convergence. In the tested scenarios, the Fiedler vectors prove to be optimal for capturing the geometry of the data. Finally, in the two cases, UPCA-W does not demonstrate effectiveness, par-ticularly in the 3D case. We leave a further investigation of this approach for future works. Additional results and visualisations are available in Supplementary Material D.

### 4.2. Clustering

In this section, we propose an unsupervised application of PW for performing clustering directly in the space of point clouds. Our approach draws inspiration from the $k$-means reformulation presented in Peyré et al. (2016) with Gromov-Wasserstein barycenters. We consider the MNIST dataset of handwritten digits with specific focus on the first five digits, from 0 to 4 (Figure 3, left). For each digit class, we consider 10 images and convert them into 2-dimensional point clouds (Figure 3, center). This results in a dataset of 50 point clouds, which we aim to cluster with respect to the digit class (thus $k = 5$). Differently from Peyré et al. (2016), we avoid applying random rotations to the dataset to highlight some benefits of PW even in scenarios where the input data are already aligned (at least in terms of reflection and rotation).

To initialize the centroids, we adopt a strategy inspired by $k$-means++ as follows. We randomly select one point cloud from the 50 and label it as the first "candidate." The first centroid is determined by applying Euclidean $k$-means clus-tering to the candidate cloud, where the number of clusters equals the number of points specified for the OT centroids (PW barycenters). This ensures that the points sampled from the candidate form a uniform representation. Next, we identify the point cloud among the remaining 49 that is the farthest from the first candidate, based on the PW distance. This farthest point cloud becomes the second "can-didate", and its centroid is computed using the same idea as for the first. By iterating this process: select the point cloud that is farthest from all previously selected candidates and compute its centroid, we obtain an initial configuration of five centroids. This approach ensures a well-distributed initialization with respect to the PW distance.

Using the same initialization technique, we compare $k$-means clustering across different OT metrics. Specifically, we present comparisons between discrete Wasserstein (Earth Mover's Distance, EMD), Gromov-Wasserstein with Eu-clidean distances (Euc-GW), with geodesic distances (Geo-GW) and PW with a Wasserstein initialization to establish

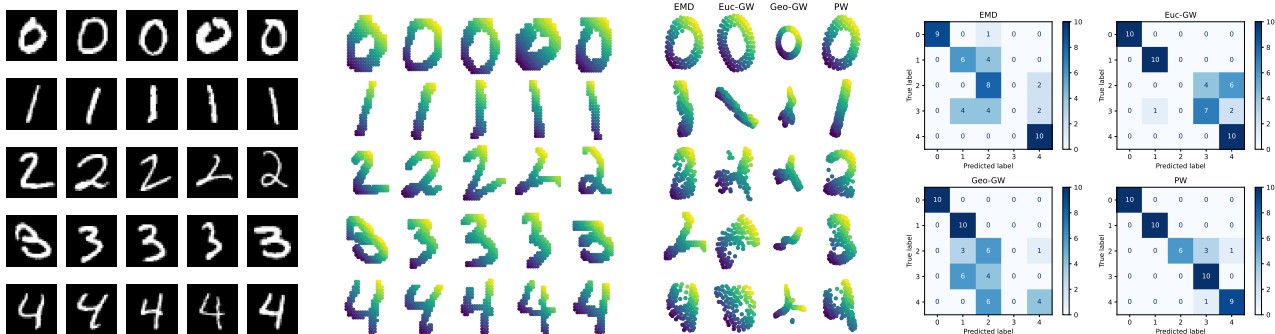

*Figure 3.* Clustering *k*-means of MNIST dataset. (Leftmost) Subset of considered images. (Center-left) Corresponding 2D point clouds representation. (Center-right) Clustering centroids computed with different OT barycenters. (Rightmost) Confusion matrices. Rows correspond to the digits, while columns correspond to the clusters. The colour is proportional to the number of digits to each cluster.

*Table 1.* Clustering results for MNIST dataset.

| ALGOROTHM | TIME (S) | ARI | NMI |
|---|---|---|---|
| EMD | **9.18** | 0.4069 | 0.5652 |
| EUC-GW | 675.19 | 0.5500 | 0.6815 |
| GEO-GW | 378.82 | 0.3797 | 0.5724 |
| PW | 130.11 | **0.7669** | **0.8361** |

initial correspondences.

The clustering results are reported in Table 1, where we provide the computational time (in seconds), the adjusted rand index (ARI) and the normalized mutual info score (NMI) for each of the presented approaches. We also provide in Figure 3 (Center-right) and Figure 3 (Rightmost) the estimated centroids (OT barycenters) and the confusion matrices, respectively. From the results, we observe that the PW-based clustering provides the best performances, in terms of ARI and NMI. Moreover, clustering results optimal for the digits 0, 1, and 3. Consistent with findings from Peyré et al. (2016), the digits where clustering is less effective are 2 and 4, reflecting greater variability in handwritten style. Differently form the GW-based clustering, PW-based successfully returns more representative centroids for all the five considered digits. EMD-based clustering proves to perform well for certain digits. However, it consistently fails to identify digit 3. The superior performance of PW over EMD underscores the significance of incorporating optimal rotations, even in scenarios where the input data are already aligned, at least in the sense that poses is consistent.

## 5. Application: tracking the morphological evolution of domestic animals

The breeding of domestic ungulates began over 9500 years ago, leading to considerable phenotypic and genetic changes,

adapted to the socio-economic and cultural requirements of human societies. In south-west Asia, from the end of the Bronze Age onwards, archaeozoological (Vila & Helmer, 2014; Vila et al., 2021; Abrahami & Michel, 2023) and palaeogenetic data (Her et al., 2022) indicate that zootechnical practices were used for the management and selection of sheep morphotypes. This led to a significant increase in phenotypic diversity and a decrease in genetic diversity. While the morphological changes observed are relatively well documented by palaeogenetic data, identifying the processes linked to morphological transformations in the bones of this species remains complex: which parts of the bone are modified (anatomical characteristics)? How do they change (bone plasticity)? Why do they change (morpho-functional adaptations linked to anthropic and environmental factors)? To track these morphological changes, traditionally archaeozoologists rely on visual comparisons and manual measurements. These methods can be time-consuming and subject to interpretational bias, especially when dealing with intraspecific variations. With the advent of 3D scanning technologies, bones can now be digitized and represented as point clouds or meshes, opening new avenues for quantitative analysis and machine learning. In this context, OT offers a mathematically robust framework to tackle the problem of comparing and interpreting bone shapes. The objective of this study is to highlight the morphological evolution over time, i.e. the transition from archaeological to modern, by directly comparing three-dimensional representations of the astragalus (ankle bone) for one archaeological sheep dated to the Chalcolithic period and one modern sheep from the same region, the Alborz mountain in Iran. With this objective, let us consider two measures $\mu_X$ and $\mu_Y$, with associated locations $X$ and $Y$ of nearly 10k vertices, representing an archaeological and a modern bone structure of the sheep species, respectively. By assigning weights $\lambda_X$ and $\lambda_Y$ respectively, we seek for a 10k PW barycenter via Algorithm 2, that defines an interpolation between the two

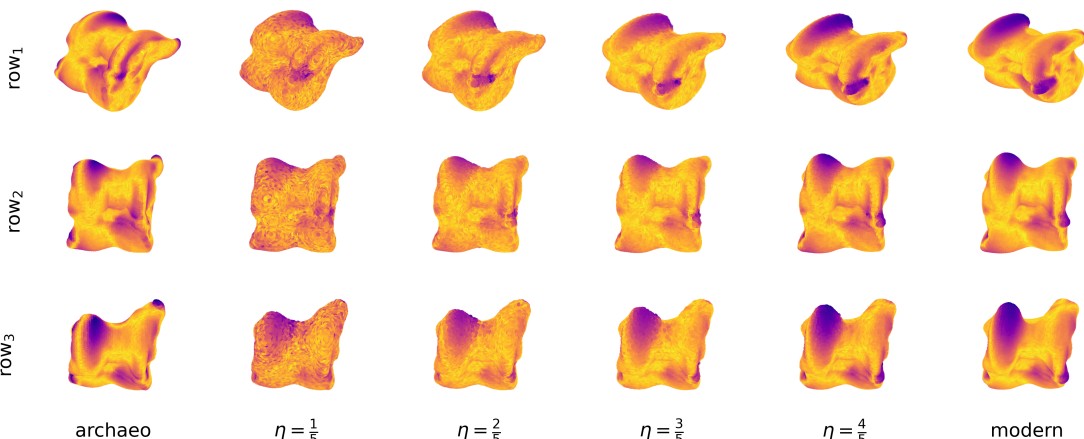

*Figure 4.* PW barycenter evolution of two 3D point clouds describing an archaeological (Leftmost) and a modern (Rightmost) astragalus of sheep's species. The four middle columns of the grid correspond to representative interpolations each assigned with a value of $\eta$. (row$_1$) Progressive interpolation in the euclidean space, note that the two input point clouds are not aligned and no priori knowledge on pairwise correspondence is considered. The $P^*$ solution of PW permits us to optimally display the frontal view (row$_2$) and top view (row$_3$), in order to match reference manuals of morphological criteria in archaeology.

bone's structures. Set $\eta \in [0, 1]$ and re-write $\lambda_X = 1 - \eta$ and $\lambda_Y = \eta$. By varying $\eta$ we can iterate the minimization problem (6) and thus model the intermediate stages of morphological changes between the two bones, enabling the study of evolutionary trajectories and species transformations over "time". In order to create a pipeline that is as robust and accurate as possible, *a priori* step in this methodology is the normalization of the data. To ensure a meaningful comparison and avoid any bias, we resort to a volume-based normalization which consists in two key steps. First, we set to the origin the volumetric center of mass. Second, we constrain the shape to have a unit volume. This technique allows the model to compute interpolations that best capture morphological changes and are less influenced by overall distortions. We remark that the purpose of this section is not in comparing different types of normalization, however, different pre-processing techniques can be tested. Figure 4 shows the evolution of the bone structure from archaeological to modern, by means of PW barycenters. In row$_1$ is reported the progressive interpolations in the 3D space. We can see that the four barycenters, corresponding to four distinct values of $\eta$, are well representative of the input point clouds. The colouring of the bones reflect the point-wise similarity between each barycenter and the modern sheep (the rightmost bone) form the archaeological sheep (the leftmost bone). In yellow are outlined the parts of the bone that are more similar, while in blue the parts that different the most. The colour of the archaeological bone, on the other hand, reflects the distance between itself and the modern bone.

As expected, we see that the first barycenter is the closest

to the archaeological bone. As we get closer to the modern sheep, the PW distance increases and the blue areas become more pronounced. By exploiting the solution of the PW barycenter problem, we benefit of a complete registration of the barycenters, we can thus visualize different views: the dorsal view (row$_2$) and the proximal view (row$_3$). These orientations facilitate the observation of changes in the overall proportions of the bone and more targeted changes, particularly to the proximal trochlea, i.e. the upper pulley-shaped articular surface. A notable observation is the widening of the lateral lip in the proximal trochlea of the modern specimen in comparison to the archaeological specimen. The proximal view also demonstrates a narrowing of the tuberculus tali and a development of the projecting medial ridge in the modern specimen compared to the archaeological specimen. For a better understanding of the bone anatomical features is provided in Figure 8 in supplementary material.

**Discussion.** The proposed approach allows us to trace the evolutionary trajectories of species by interpolating between bone shapes. This method provides archaeozoologists with a powerful quantitative tool to infer how species adapted, evolved, or were selectively bred by humans. Furthermore, the same technique could be used to compute a "mean" representative bone shape for species for which morphological criteria are not well-defined. This would reinforce and supplement studies combining machine learning and archaeozoology to identify morphologically related taxa (Miele et al., 2020; Moclán et al., 2023; Vuillien et al., 2025). In conclusion, by directly working on 3D models, this approach offers a robust solution that aim to avoid the

subjectivity inherent in traditional morphological analysis. The use of the PW distance and PW barycenters introduce a rigorous and quantitative framework for analyzing shapes while offering to archaeologists a detailed and objective tool for interpreting species evolution, domestication patterns, and morphological diversity, ultimately enhancing our understanding of the past.

## 6. Conclusions

In this paper, we carefully defined a space of discrete probability measures over which Procrustes-Wasserstein is a distance and provided a formal proof of such claim. This opens the door to a wider application of PW in various machine learning tasks, particularly when dealing with complex data structures. We also introduced PW barycenters extending the literature of OT barycenters. Our formulation enables the construction of representative measures that exhibit an improved visual loyalty to the geometry of the observed data. We propose applications that demonstrated the properties and advantages of our approach, with comparisons with state-of-the-art methods. Future works could explore denser formulations of the barycenter problem (via entropic regularisation) leading to smoother solutions and broader applicability to large-scale datasets.

## Acknowledgements

The work has received financial support from the CNRS through the MITI interdisciplinary programs and the Junior Professor Chair (Chaire de Professeur Junior, CPJ) funded by the French National Research Agency (ANR). We would like to thank Arch'AI'Story project (Ministère de l'Enseignement Supérieur et de la Recherche and University Côte d'Azur) for funding this project. We would like to express our gratitude to Dr. Hossein Davoudi (Bioarchaeology Laboratory Central Laboratory, University of Tehran, Iran) and Dr. Marjan Mashkour (Bioarchaeology, Interactions societies-environments laboratory-UMR 7209, CNRS, National Museum of Natural History of Paris, France) for their support and authorization to provide samples of the modern and archaeological sheep. We also wish to warmly thank Cédric Vincent-Cuaz for the enlightening discussions we had with him around this work as well as for sharing with us his point of view regarding the Procrustes-Wasserstein distance. Our gratitude is finally extended to the anonymous reviewers whose contributions proved to be of substantial value in enhancing the quality of the article.

## Impact Statement

This paper presents work whose goal is to advance the field of Machine Learning. There are many potential societal consequences of our work, none which we feel must be specifically highlighted here.

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

# A. Proof of Theorem 2.1.

*Proof.* First we check that $PW_2(\mu_X, \mu_Y) = 0$ iff $\mu_X \sim \mu_Y$. The left implication is clear : if $\mu_X \sim \mu_Y$, it means that there is $(\sigma^*, P^*)$ such that $Y = \sigma^*(X)P^*$ and $\mathbf{q} = \sigma^*(\mathbf{p})$. Then $(\sigma^*, P^*)$ is the solution of the problem in Eq. (5), with the Kantorovich formulation being equivalent to the Monge's one. Vice-versa, if

$$PW_2(\mu_X, \mu_Y) = \min_{P \in \mathcal{O}(d)} W_2(\mu_X, \mu_{YP}) = 0,$$

it means that there exists à $P^*$ such that the 2-Wasserstein distance between $\mu_X$ and $\mu_{YP^*}$ is null, or equivalently that $\mu_X$ and $\mu_{YP}$ are the same measure up to a permutation of the points in the support together with their masses.

Second we prove that $PW_2(\mu_X, \mu_Y) = PW_2(\mu_Y, \mu_X)$ for all $\mu_X, \mu_Y$ in $\mathcal{M}_d$. Let us assume that $(P^*, \Gamma^*)$ is solution of Problem (4), with $P^* \in \mathcal{O}(d)$ and $\Gamma^* \in \Pi(\mathbf{p}, \mathbf{q})$. Then $d(\mu_Y, \mu_X)$ is defined by the maximization over $Q \in \mathcal{O}(d)$ and $\Theta \in \Pi(\mathbf{q}, \mathbf{p})$ of

$$\begin{aligned}
\langle YQ^T X^T, \Theta \rangle_F = \mathrm{tr}(\Theta^T Y Q^T X^T) &= \mathrm{tr}(XQY^T \Theta) \\
&= \mathrm{tr}(\Theta X Q Y^T) = \langle XQY^T, \Theta^T \rangle_F \\
&\leq \langle X(P^*)^T Y^T, \Gamma^* \rangle_F,
\end{aligned}$$

by optimality of $(P^*, \Gamma^*)$ and where we used that the trace of a matrix equals the trace of its transposed and the trace is invariant under cyclic permutations of its arguments. The above equation shows that if $(P^*, \Gamma^*)$ is the stationary point leading to $PW_2(\mu_X, \mu_Y)$ then, $\left((P^*)^T (\Gamma^*)^T\right)$ is the solution leading to $PW_2(\mu_Y, \mu_X)$ and vice-versa. Thanks to Eq. (3), it is now immediate to verify that $PW_2(\mu_X, \mu_Y) = PW_2(\mu_Y, \mu_X)$.

Third, we show that the triangular inequality is satisfied : $PW_2(\mu_X, \mu_Y) \leq PW_2(\mu_X, \mu_Z) + PW_2(\mu_Z, \mu_Y)$, for all $\mu_X, \mu_Y, \mu_Z$ in $\mathcal{M}(d)$. For all $\mu_Z \in \mathcal{M}_D$ it holds that

$$W_2(\mu_X, \mu_{YP}) \leq W_2(\mu_X, \mu_{ZP}) + W_2(\mu_{YP}, \mu_{ZP}) = W_2(\mu_X, \mu_{ZP}) + W_2(\mu_Y, \mu_Z),$$

where the first inequality holds since the Wasserstein distance is symmetric and satisfies the triangular inequality (as any distance) and the second equality comes from the fact that if we equally rotate or reflect the supports of two measures the Euclidean distances between any pair of points in the supports will be unchanged. From the above equation, paired with Eq. (2) we deduce that, for any $\mu_Z \in \mathcal{M}_d$

$$PW_2(\mu_X, \mu_Y) \leq PW_2(\mu_X, \mu_Z) + W_2(\mu_Y, \mu_Z).$$

Now, if we replace $Z$ with $Z^* = ZP^*$ where $P^*$ is solution of

$$\min_{P \in \mathcal{O}(d)} W_2(\mu_Y, \mu_{ZP}),$$

we obtain

$$PW_2(\mu_X, \mu_Y) \leq PW_2(\mu_X, \mu_{Z^*}) + W_2(\mu_Y, \mu_{Z^*}) = PW_2(\mu_X, \mu_{Z^*}) + PW_2(\mu_Y, \mu_Z),$$

where the last equality holds by the definition of PW. Finally, since $Z$ and $Z^*$ only differ by right-multiplication with an orthogonal matrix, $PW_2(\mu_X, \mu_{Z^*}) = PW_2(\mu_X, \mu_Z)$. $\qquad\square$

## B. Optimization of $PW_2$ with respect to p

In this section we provide for clarity the algorithm for the optimization of the weights **p**, already proposed by Cuturi & Doucet (2014) within the context of Wasserstein barycenters. In our framework we assume $\mathbf{p} \in \Sigma_n$ and denote $\circ$ the Schur's product.

---

**Algorithm 3** Optimization of **p**

1: **Input:** Cost matrices with orthogonal alignments $C_{P_j} \in \mathbb{R}^{n \times n_j}$ and histograms $\mathbf{p}_j \in \mathbb{R}^{n_j}$ for $j = 1, \ldots, r$
2: Set $\hat{p} = \tilde{p} = \mathbb{1}_n/n$
3: **while** not converged **do**
4:   $\beta = (t+1)/2$
5:   $\mathbf{p} \leftarrow (1 - \beta^{-1})\hat{p} + \beta^{-1}\tilde{p}$
6:   $\alpha \leftarrow \frac{1}{r}\sum_{j=1}^r \alpha_j^*$ using all dual optima $\alpha_j^*$ of Eq. (9)
7:   $\tilde{p} \leftarrow \tilde{p} \circ e^{-t_0\beta\alpha}$
8:   $\tilde{p} \leftarrow \tilde{p}/\tilde{p}^T\mathbb{1}_n$
9:   $\hat{p} \leftarrow (1 - \beta^{-1})\hat{p} + \beta^{-1}\tilde{p}$
10:   $t \leftarrow t + 1$
11: **end while**
12: **Return:** **p**

---

## C. Regularized PW barycenter(s)

As common in the OT community, we extend in this section the PW barycenter problem by adding an entropic regularization. Consider the following relaxed version of the PW problem

$$PW_{\epsilon,2}^2(\mu_X, \mu_{YP}) = \min_{\substack{P \in \mathcal{O}(d) \\ \Gamma \in \Pi(\mathbf{p},\mathbf{q})}} \langle C_P(X,Y), \Gamma \rangle_F + \epsilon H(\Gamma), \tag{10}$$

where $\epsilon$ is a non-negative parameter controlling the strength of the regularization and $H(\Gamma) := \sum_{i,j} \Gamma_{ij} \log(\Gamma_{ij})$ is the entropy.

Looking for a measure $\mu_X$ with unknown support $X \in \mathbb{R}^{n \times d}$ and weights **p**, given $r$ measures $\mu_{X_1}, \ldots, \mu_{X_r}$, translates into the following minimization problem

$$f_\epsilon(\mathbf{p}, X) := \sum_{j=1}^r \lambda_j PW_{\epsilon,2}^2(\mu_X, \mu_{X_j}). \tag{11}$$

The resolution of this problem can be done similarly as for the classical case. Given the probability vector **p** and assuming $(\Gamma_j^*, P_j^*)$ are the solutions of the regularized PW problem $PW_{\epsilon,2}^2(\mu_X, \mu_{X_j})$, the Newton update is still given by Eq. (8). Notably, at the optimum, both the gradient and the Hessian of Eq. (11) are independent of the regularization term $H(\cdot)$. The optimization scheme for the computation of the barycenter is equivalent to Algorithm 2, where in this case we solve each PW sub-problem independently using the Sinkhorn algorithm Cuturi (2013). Under this framework, the optimization of Eq. (11) can be seen as a particular case of the one discussed in Alvarez-Melis et al. (2019).

To illustrate the effectiveness of the proposed barycenter, we present a 2D toy example where we consider two measures $\mu_X$ and $\mu_Y$, with associated locations $X$ and $Y$, representing different instances of the "same" object. Specifically, $X$ represents a point cloud depicting a bird in a particular pose. We define $Y$ as a modified version of $X$ with the following transformations: addition of Gaussian perturbations and random vertex permutation; addition of extra vertices; application of a random rotation. Within this setting, our goal is to generate interpolated shapes that progressively move from $X$ to $Y$. Set thus $\eta \in [0,1]$ and re-write $\lambda_X = 1 - \eta$ and $\lambda_Y = \eta$. By varying $\eta$ we can iterate the minimization problem and model the intermediate stages. We compute PW barycenters using both the classical problem formulation (Problem 6) and the regularized version (Problem 11).

In Figure 5 we can follow the progressive interpolations between the two point clouds. The four central columns correspond to PW barycenters at different interpolation steps, while the two rows compare the classical PW barycenter (top) with the

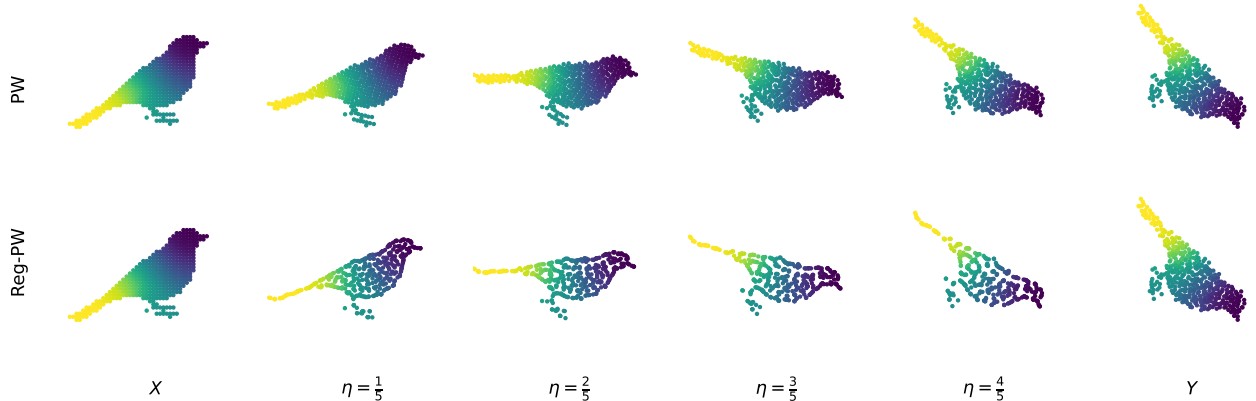

*Figure 5.* Progressive interpolations of two point clouds describing a bird in different positions. The four central columns of the grid correspond to PW barycenters, each reflecting a specific interpolation step (defined by $\eta$). The two rows represent barycenters calculated using the classical formulation (PW) and considering an entropic relaxation of the problem (Reg-PW).

relaxed version (bottom). The colors of the barycenters are given by transporting the colors of $X$ via the optimal plan $\Gamma_X^*$. This highlights how the cloud evolves while maintaining the structural consistency of the original shape.

# D. Additional results

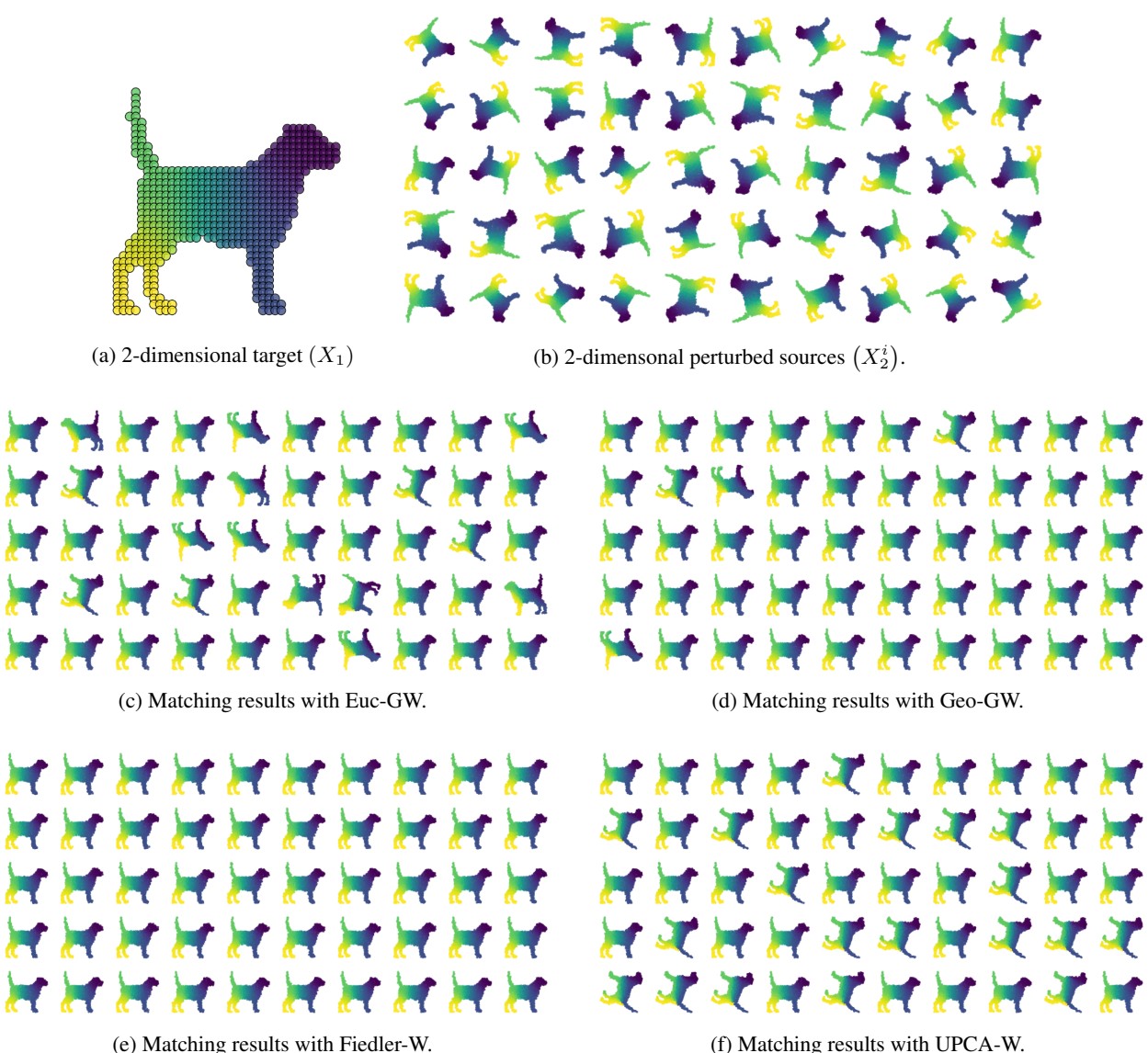

(a) 2-dimensional target $(X_1)$

(b) 2-dimensonal perturbed sources $(X_2^i)$.

(c) Matching results with Euc-GW.

(d) Matching results with Geo-GW.

(e) Matching results with Fiedler-W.

(f) Matching results with UPCA-W.

*Figure 6.* Comparisons of PW matching results with different initialization approaches (2D dog). Matchings reflect the convergence results reported in Figure 2

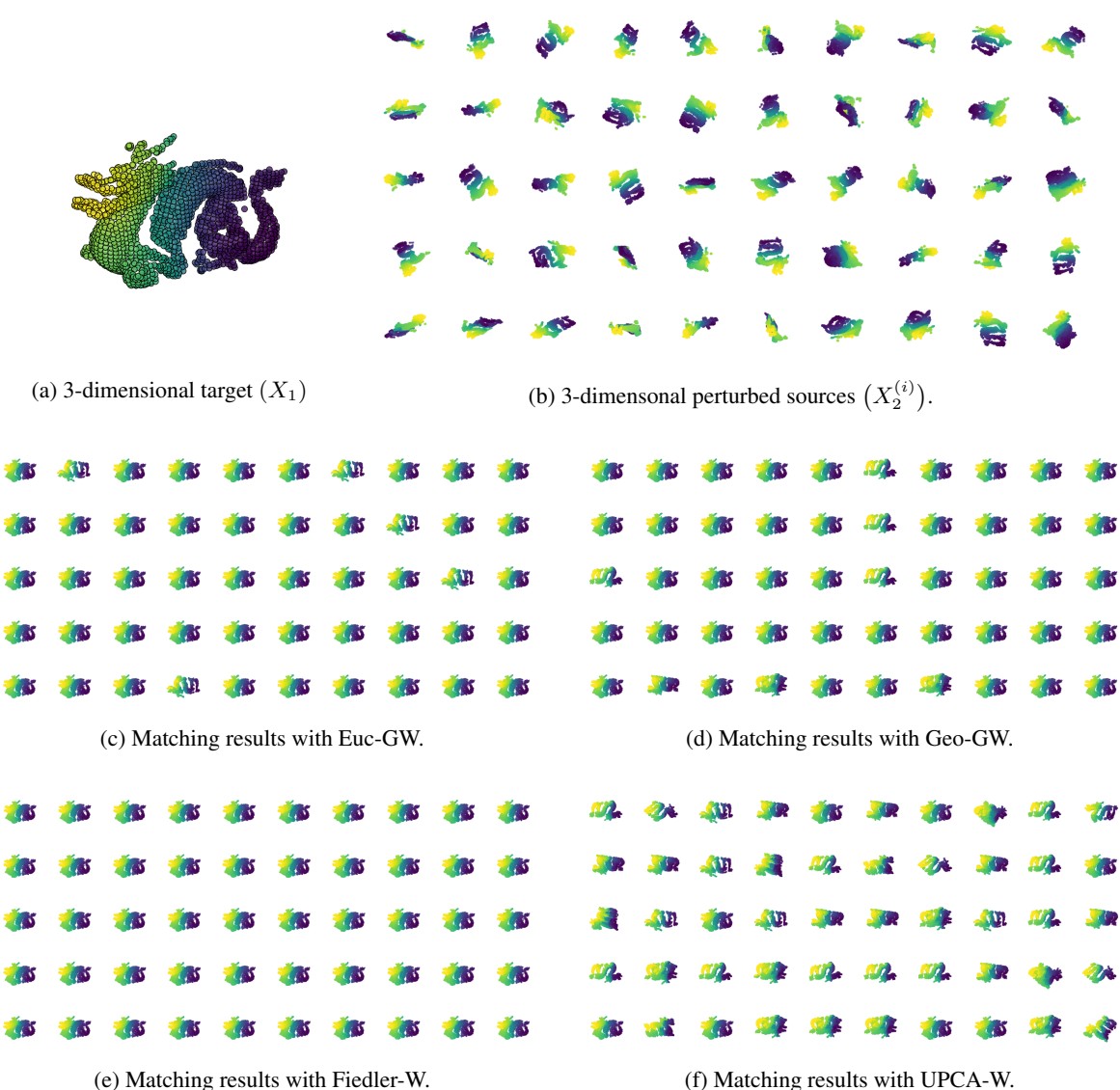

(a) 3-dimensional target $(X_1)$

(b) 3-dimensonal perturbed sources $\left(X_2^{(i)}\right)$.

(c) Matching results with Euc-GW.

(d) Matching results with Geo-GW.

(e) Matching results with Fiedler-W.

(f) Matching results with UPCA-W.

*Figure 7.* Comparisons of PW matching results with different initialization approaches (3D dragon). Matchings reflect the convergence results reported in Figure 2

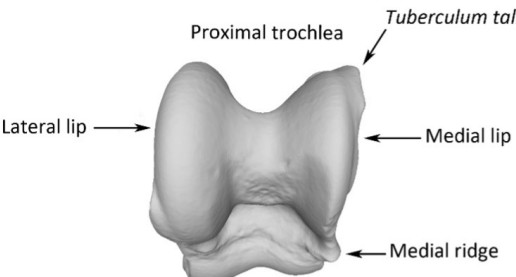

*Figure 8.* Proximal view of the modern astragalus and its main anatomical features presented in the Section 5

