# OpenReview forum: "An in depth look at the Procrustes-Wasserstein distance: properties and barycenters"
_ICML.cc/2025/Conference — ICML 2025 poster_

### Official Review · Reviewer_HyoR · 2025-03-11

**Overall Recommendation:** 3

**Summary:**

This paper defines a quotient space of discrtete measures over which PW is a distance and provides an estimation algorithm for the PW barycenter. This paper then shows that one of the main advantages of PW is its capability to produce very faithful barycenters in particular conditions. Experiments have verified its effectiveness in point cloud alignment and shape analysis, and its application in archaeological data shows the practical significance of the proposed method.


## update after rebuttal
I appreciate the response provided in the rebuttal, which addresses some of my concerns, and I maintain the score.

**Claims And Evidence:**

yes

**Essential References Not Discussed:**

No

**Experimental Designs Or Analyses:**

The archaeological application experiment only shows the interpolation results. Are there some quantitative indicators to evaluate the results?

**Methods And Evaluation Criteria:**

yes

**Other Comments Or Suggestions:**

As above

**Other Strengths And Weaknesses:**

Strengths
1. This paper proves the distance property of PW distance on discrete measure quotient space.
2. This paper introduces the notion of PW barycenter.

Weaknesses
1. Figure 1 lacks more explanation. The caption shows: OT barycenters using (a) Free Wasserstein (b) Gromov-Wasserstein with MDS (c) Gromov-Wasserstein with TSNE. However, (a-c) is not introduced in the paper. In addition, the caption should introduce what task this is, such as a matching task or another.
2. MDS and TSNE only have abbreviations, no references or full names are provided.
3. The archaeological application experiment only shows the interpolation results. Are there some quantitative indicators to evaluate the results?

**Questions For Authors:**

As above

**Relation To Broader Scientific Literature:**

Procrustes Wasserstein (PW) was introduced in the literature as an optimal transport (OT) distance, alternative to Wasserstein and more suited to tasks such as the alignment and comparison of point clouds. This paper builds a space of discrete probability measures and shows that over that space PW actually is a distance.

**Theoretical Claims:**

Yes, I don’t find any issues with the proofs for theoretical claims.

---

> ### Author Rebuttal · Authors · 2025-03-31
>
> First of all, thanks for the kind remarks and for reviewing the paper and checking the supplementary. We provide in the following the answer to the specific questions.
>
> > 1-2. Figure 1 lacks more explanation. The caption shows: OT barycenters using (a) Free Wasserstein (b) Gromov-Wasserstein with MDS (c) Gromov-Wasserstein with TSNE. However, (a-c) is not introduced in the paper. In addition, the caption should introduce what task this is, such as a matching task or another. MDS and TSNE only have abbreviations, no references or full names are provided.
>
> We apologize for the lack of clarity regarding Figure 1. Below, we provide additional details about the experiment and include references to the MDS and TSNE methods, which were indeed missing from the text. Here is the new caption for Figure 1, which will be updated in the final version of the paper:
>
> “(Top) Two point clouds representing a bird shape in different positions. OT barycenters using (a) Exact Free Wasserstein [1] (b) Gromov-Wasserstein [2] with MDS [3] (c) Gromov-Wasserstein with TSNE [4] (d) Procrustes-Wasserstein.”
>
> In this motivational example, we compare different OT techniques for computing barycenters. The two point clouds at the top represent the input measures (depicting a bird shape, without and with noise, and in different positions). The techniques used in panels (a-d) are, respectively: exact Wasserstein barycenters [1], Gromov-Wasserstein barycenters [2] with two different visualization techniques, Multidimensional Scaling (MDS, [3]) and t-distributed Stochastic Neighbor Embedding (TSNE, [4]), and finally, the proposed PW method. The results highlight how the PW barycenter provides the clearest and most precise representation, even for local geometric structures. Additionally, the barycenters are colored by transporting the color of the first bird (top left) onto the computed barycenter.
>
> > 3. The archaeological application experiment only shows the interpolation results. Are there some quantitative indicators to evaluate the results?
>
> Thank you for this central question. Quantitative methods are available for the quantification of morphological deformations based on simplified morphological patterns in archaeology (e.g. geometric morphometric methods). However, no quantitative method exists that can be used with the entire 3D bone. Ensuring that the observed morphological deformations on the barycentre correspond to a biological reality based on archaeological data is a significant challenge. While acknowledging these challenges, we firmly believe that this preliminary approach signifies a significant opportunity for interdisciplinary research.
>
> References:
>
> [1] Cuturi, Marco, and Arnaud Doucet. "Fast computation of Wasserstein barycenters." International conference on machine learning. PMLR, 2014.
>
> [2] Peyré, Gabriel, Marco Cuturi, and Justin Solomon. "Gromov-wasserstein averaging of kernel and distance matrices." International conference on machine learning. PMLR, 2016.
>
> [3] Borg, Ingwer, and Patrick JF Groenen. Modern multidimensional scaling: Theory and applications. Springer Science & Business Media, 2007.
>
> [4] Maaten, Laurens van der, and Geoffrey Hinton. "Visualizing data using t-SNE." Journal of machine learning research 9.Nov (2008): 2579-2605.
>
> [5] Cuturi, Marco. "Sinkhorn distances: Lightspeed computation of optimal transport." Advances in neural information processing systems 26 (2013).

---

### Official Review · Reviewer_GNtQ · 2025-03-11

**Overall Recommendation:** 3

**Summary:**

The paper proposes a method for aligning and matching point clouds based on Optimal Transport using the Procrustes-Wasserstein distance (PW), allowing the alignment to be calculated by also taking pose transformations into account.

The paper shows that PW is indeed a distance in the space of discrete probability measures. It also introduces an algorithm to calculate the barycentres of the measures with respect to PW. The contribution is validated by showing how to initialise optimal transport, testing four main approaches on a synthetic dataset. The method is then tested for clustering on a subset of MNIST and compared with other Optimal Transport metrics.   A final application shows how the method was crucial for tracing the morphological evolution of sheep based on astragalus alignments from the Chalcolithic period up to the modern age.

**Claims And Evidence:**

The claims of the article are:

1) Using PW, it is possible to encompass geometric transformations. Other articles have already proved this claim (we discuss this in another section).

 2) The Procrustes-Wasserstein distance is a distance on the space of discrete probability measures, and this claim is proven in the supplementary.
This claim is used to define the barycentres.

3) The calculation of the barycentres is relevant for obtaining clusters that take geometric transformations into account.
The algorithm is as precise as the derivations that lead to its optimisation.

I'm unsure if it is the first algorithm (as claimed in the introduction). The Frechet mean can actually be considered a barycentre of a discrete measure, and its computation was introduced by Zemel et *Al.* [1], referring to the OT (the authors do not cite the article).

The last claim concerns the initialisation of $\Gamma_{0}$ or admissible transport plans.

From the two pictures in Figure 2 and the comments in the text, it appears that only the Fiedler vector ensures convergence. However, it should be noted that the Fiedler vector only works if the Laplacian matrix is noise-free, as in the case studied, which is defined by a synthetic set.

Thus, the problem of the initialisation of $\Gamma\_{0}$ remains open for real cases.

[1] Zemel, Y. and Panaretos, V.M., 2019. *Fréchet means and Procrustes analysis in Wasserstein space.*

**Essential References Not Discussed:**

Essential references not discussed have been highlighted in the previous paragraph.

**Experimental Designs Or Analyses:**

As specified above, I am critical about the initialisation of $\Gamma_{0}$.

As shown in the paper, only with the Fiedler vector is convergence obtained at the minimum distance detecting $\Gamma_{0}$.  However, the test was carried out on synthetic data and, thus, without noise.  In the presence of noise, the Fiedler Vector does not separate the components correctly.

Furthermore,  the presupposition on the space of measures is very strong. As noted in Zemel [1] an optimal transport map may fail to exist, and instead, one may need to solve the relaxed Monge problem.

**Methods And Evaluation Criteria:**

Regarding the experiments, the article follows the literature somewhat (see, for example, Cuturi and Doucet) with no tables (here, only Table 1 to compare with other OT metrics) and only pictures without showing measures of similarities and failure values. Even for the main application, there is only one image.

However, this is partly justified by the fact that the pictures serve as proof of concept, and the work is not experimental but more theory-oriented.

**Other Comments Or Suggestions:**

Some spelling errors (line:column)


“062:1  the repeated twice"

“96:2   discrtete”

“112:1  matchin”

“113:   applicaton”

“123:1  the the”

“210:2 interative”

“289:1  space.Morover”

“325:2 calstering”

“325:2 provide”

“348:1 cluetsring”

**Other Strengths And Weaknesses:**

The article makes a simple and useful contribution but has to be compared with the works mentioned above.

It has many limitations:
1. It is not well written; there are many spelling mistakes, which I point out in the next paragraph.

2. It is limited in its experiments. Perhaps it should have reduced the space for describing the application and compared the results better with the literature, see, for example, [3].

3. The initialisation of $\Gamma0_{0}$ should have been analysed more deeply and with real examples.

[3] Jin, K., Liu, C., and Xia, C. *Two-sided Wasserstein Procrustes analysis.* In IJCAI, pp. 3515–3521, 2021. (this paper was cited)

**Questions For Authors:**

Please look into both Even's and Zemel's papers.

You never mention algorithms for computing OT.

For example, Cuturi & Doucet  refer, in their Algorithm 3 Cuturi fast algorithm [Cuturi 2013]

**Relation To Broader Scientific Literature:**

The article is well placed, it is definitely a contribution.

However, the paper *Aligning Embeddings and Geometric Random Graphs: Informational Results and Computational Approaches for the Procrustes-Wasserstein Problem' by Even et Al., appeared at Neurips 2024, which deals with the problem in a very general way.

Although the authors limit their method to discrete measures, they should have mentioned this and, therefore, related their findings to the more general ones of Even et *Al.* [2].
Indeed, this should be done in the revised version.

Another limitation is perhaps not looking at the Frechet mean (e.g. Zemel's [1]).

[2] Even, M., Ganassali, L., Maier, J. and Massoulié, L., 2024. *Aligning Embeddings and Geometric Random Graphs: Informational Results and Computational Approaches for the Procrustes-Wasserstein Problem*

**Theoretical Claims:**

I checked.

---

> ### Author Rebuttal · Authors · 2025-03-31
>
> First of all, thank you very much for the careful reading of the paper, as well as for the analysis and comments, they are truly appreciated. We have addressed all the grammatical errors identified in the text, and the corrections will be included in the final version of the paper. We also appreciate the suggestions regarding [Zemel et al., 2019] and [Even et al., 2024], which are now references in the main paper. However, while both studies align with the general problems discussed in our paper, they differ from the specific contributions presented in our work. Below, we provide a detailed discussion on the connections between our work and these references, along with responses to the specific points raised in the review.
>
> > 1. Relation to [Zemel et al., 2019]:
>
> It is true that the barycenter problem is, by definition, equivalent to computing the Fréchet mean. However, the work of [Zemel et al., 2019] aims to highlight the analogy between the Fréchet mean computation, i.e. barycenters, computed in the Wasserstein space and Procrustes analysis as methods for the “registration” of distributions. Unlike our work, [Zemel et al., 2019] does not minimize a Procrustes-Wasserstein cost, but rather a classical Wasserstein one. Specifically, is it worth to note that Algorithm 1 in [Zemel et al., 2019] consists of N pairwise OT computations, followed by a barycenter update performed directly in the Wasserstein space. We apologize if this distinction was not sufficiently clear in our manuscript. Our work directly addresses the PW problem, introducing the first formalization of PW barycenters along with a practical and efficient algorithm for their computation (Alg. 2). This approach is fundamentally different from [Zemel et al., 2019].
>
> > 2. Relation to [Even et al., 2024]:
>
> The problem addressed in [Even et al., 2024] tackles the matching of pairs of distributions from a theoretical perspective, providing thorough results about the convergence of their ML estimates. However, it represents a special case of the more general Procrustes-Wasserstein problem which is approached in the current work. Indeed [Even et al., 2024] specifically considers distributions (pairs of point clouds) that have the same number of points; in their model, the signal only differs from the dependent point cloud by a permutation of the points plus an isometry, then Gaussian noise is added to each point. The OT problem they address corresponds thus to the Monge formulation of OT. In contrast, the problem approached in the current work generalizes to cases where the distributions may have different numbers of points (as can be also observed in the experiments we propose) thereby corresponding to the more general Kantorovich formulation of OT. We look for doubly stochastic solutions (i.e. transport plans), they look for permutation matrices.
>
> > 3. Initialization of $\Gamma_0$
>
> We agree with you, the initialization of the Procrustes-Wasserstein matching in real-world scenarios remains an open challenge and addressing it is not the primary goal of our work.
> However, it is important to note that, differently from what you state, in the matching problem we tackled, noise is present in the Laplacian (although at a level such that the global geometric structure of the shape remains visible). As detailed in our section, the generated clouds $X^i_2$ (meant to be matched to the pivot $X$) also include Gaussian noise applied to each point of the pivot cloud and thus represent a perturbation of the pivot thus adding noise to the corresponding Laplacian matrices. We emphasize that the perturbations applied to the clouds $X^i_2$ are designed to simulate real data scenarios. However, the Wasserstein distance between Fiedler vectors can still find a good and sufficient matching, which can then be used as the initial $\Gamma_0$​ for solving Alg. 1.
>
> > 4. “You never mention algorithms for computing OT.”
>
> We apologize for any lack of clarity in the text. Please refer to responses 2 and 3 to reviewer 6RWS for details.
>
> References :
>
> [1] Cuturi, Marco, and Arnaud Doucet. "Fast computation of Wasserstein barycenters." International conference on machine learning. PMLR, 2014.
>
> [2] Peyré, Gabriel, Marco Cuturi, and Justin Solomon. "Gromov-wasserstein averaging of kernel and distance matrices." International conference on machine learning. PMLR, 2016.
>
> [3] Borg, Ingwer, and Patrick JF Groenen. Modern multidimensional scaling: Theory and applications. Springer Science & Business Media, 2007.
>
> [4] Maaten, Laurens van der, and Geoffrey Hinton. "Visualizing data using t-SNE." Journal of machine learning research 9.Nov (2008): 2579-2605.
>
> [5] Cuturi, Marco. "Sinkhorn distances: Lightspeed computation of optimal transport." Advances in neural information processing systems 26 (2013).

---

### Official Review · Reviewer_6RWS · 2025-03-13

**Overall Recommendation:** 3

**Summary:**

This paper extends the Wasserstein barycenter to the Procrustes-Wasserstein (PW) barycenter, offering a novel approach to computing representative shapes from a collection of point clouds. Additionally, it provides a proof that the Procrustes-Wasserstein distance satisfies the properties of a valid distance metric.

**Claims And Evidence:**

The claims made in the submission are supported by clear and convincing evidence.

**Essential References Not Discussed:**

No

**Ethics Expertise Needed:**

["Other expertise"]

**Experimental Designs Or Analyses:**

Yes, it is soundness.

**Methods And Evaluation Criteria:**

Yes

**Other Comments Or Suggestions:**

None

**Other Strengths And Weaknesses:**

The paper does not provide any convergence proof for the Procrustes-Wasserstein barycenter, which raises concerns about the theoretical guarantees of the proposed method. Additionally, it lacks an analysis of the computational complexity, which is crucial for understanding the feasibility and scalability of the approach. A detailed complexity analysis would help clarify the efficiency of the algorithm, especially in large-scale point cloud applications.

**Questions For Authors:**

1. What is the convergence behavior of the proposed Procrustes-Wasserstein (PW) barycenter?
2. How does the performance change when replacing EMD with the Sinkhorn algorithm?
3. How does the computational complexity of PW barycenters compare to Gromov-Wasserstein (GW) barycenters in theory?
4. How about the performance in partial overlapping shape matching

**Relation To Broader Scientific Literature:**

Procrustes-Wasserstein (PW) barycenter is simple and interesting.

**Theoretical Claims:**

I have checked the correctness that the Procrustes-Wasserstein distance satisfies the properties of a metric.

---

> ### Author Rebuttal · Authors · 2025-03-31
>
> Thanks for your kind remarks as well as for reviewing the paper and checking the correctness of the main proof.
>
> > 1. What is the convergence behavior of the proposed Procrustes-Wasserstein (PW) barycenter?
>
> Inspecting the convergence properties of the PW barycenters is a crucial homework and a very important topic. We are currently working on it, but we honestly believe that having introduced the PW barycenters might be enough for one paper. Moreover, we respectfully point out that the convergence behavior of OT barycenters is not reported in related papers such as (for instance) [1] and [2], both accepted at this conference.
>
> > 2. How does the performance change when replacing EMD with the Sinkhorn algorithm?
>
> The relevance of the proposed method for computing the barycenter also lies in the quality of the obtained solution, specifically in terms of local geometric details. However, as common in OT, an entropic penalisation can be introduced in the objective function to stabilize the solution and speed up computations. To this end, we conducted new experiments adopting the Sinkhorn algorithm in place of EMD for the pairwise OT computations (line 8 of Alg. 1) and denoted the corresponding barycenter method as Procrustes-Sinkhorn barycenter (PSb). In the camera ready version of the paper, we might dedicate an additional section to provide the formal definition of the regularized problem and the derivation of the update formulas. We exactly observed what is known from the literature: the entropic penalty can be used to stabilize the solution and speed up computations but it comes at a price. A higher $\epsilon$ (the strength of the penalty term) is required to have a faster convergence (see Table here after) but leads to less accurate, deformed barycenters. However, for ML pipelines where there is no need for such a high level of barycentric detail, PSb certainly represents a valuable option.
>
> |   |      PWb with EMD (Alg. 1)      |  PSb ($\epsilon=10$) |  PSb  ($\epsilon=1$) | PSb ($\epsilon=0.01$) |
> |----------|:-------------:|:------------:|:------------:|:------------:|
> | Mean time & std. (s) |  1.91 ± 0.14 | 1.18 ± 0.26 | 5.69 ± 1.73 | 13.74 ± 1.12
>
> Results over 10 barycenter computations (10 runs) considering two input measures of nearly 500 points (as in Figure 1).
>
> > 3. How does the computational complexity of PW barycenters compare to Gromov-Wasserstein (GW) barycenters in theory?
>
> Thanks for this question. In practical applications, the PW algorithm (Alg. 1) is dominated by the OT computation. Since the Procrustes update has an explicit formula and the dimension D is small (in our case 2 or 3), the cost of the Procrustes step is negligible. Thus, the main computational bottleneck remains the EMD computations, which scales as O(n3 log(n)) (or O(n^2) if one adopts Sinkhorn divergences) according to [3].  The algorithmic complexity of computing PW is thus O(Kn3(log(n)), where K is the number of iterations between the Wasserstein and the Procrustes optimisation (also the number of Ping-Pong steps in [Even et al. 24], with reference to answer to Review GNtQ). By direct analogy with Alg. 2 in [1], the computational complexity of the PW barycenter algorithm follows the same structure as that of Wasserstein barycenters, with the key difference that each Wasserstein update is replaced by a PW computation. The theoretical complexity of a PW barycenter in Alg. 2 is O(TNKn3log(n)) (respectively O(TNKn^2) with Sinkhorn divergences) where N is the number of input measures, n is the maximal number of points in the input clouds and T is the number of barycenter’s updates (line 7, Alg. 2) which are essentially equivalent to the ones of a Wassersetin barycenter and so negligible with respect to the PW updates. Noting that in all our experiments we set the maximum value of K to 100 and assuming this value as a constant << N, which is reasonable when working with real point clouds (thousands of points) the computational complexity of  a PW barycenter is the same as the one of  a Wassesrtain barycenter, obviously up to a constant.  That complexity is smaller than those of GW barycenters  (the simple evaluation of the GW$_2$ cost function, without considering the network flow algorithms to solve it, is at least cubic).
>
>
> References:
>
> [1] Cuturi, Marco, and Arnaud Doucet. "Fast computation of Wasserstein barycenters." International conference on machine learning. PMLR, 2014.
>
> [2] Peyré, Gabriel, Marco Cuturi, and Justin Solomon. "Gromov-wasserstein averaging of kernel and distance matrices." International conference on machine learning. PMLR, 2016.
>
> [3] Cuturi, Marco. "Sinkhorn distances: Lightspeed computation of optimal transport." Advances in neural information processing systems 26 (2013).

---

### Official Review · Reviewer_nX3Q · 2025-03-14

**Overall Recommendation:** 1

**Summary:**

The authors propose that in the quotient space of the discrete measures over the rigid transformation equivalence, the Procrustes Wasserstein is a metric. To calculate the Procrustes Wasserstein distance, they introduce several initialization methods, i.e. Euc-GW, Geo-GW, Fiedler-W and UPCA-W. The barycenter problem and algorithms are also provided.

**Claims And Evidence:**

The claims are supported by evidence.

**Essential References Not Discussed:**

N/A

**Experimental Designs Or Analyses:**

Details of the experimental setup are provided. It would be better if the authors could provide the comparisons with baselines for morphological evolution applications, especially with the Gromov-Wasserstein methods.

**Methods And Evaluation Criteria:**

Although the authors show the application of barycenter in tracking the morphological evolution of domestic animals, no comparisons are made with other existing methods, can you show the results by the baselines for morphological interpolations?

**Other Comments Or Suggestions:**

N/A

**Other Strengths And Weaknesses:**

My main concern is the novelty of this work. The quotient space and metric properties are very incremental, and the methods for calculating the initial plan are based on previous methods.

**Questions For Authors:**

In Figure 1, can the authors comment on the gromov-Wasserestein results? Why would there be some outlier points?

**Relation To Broader Scientific Literature:**

The proposed method could be applied to point cloud analysis where rigid transformations are prevalent but should not affect the comparisons.

**Theoretical Claims:**

The proof for Theorem 2.1 and Corollary 2.2 are correct.

---

> ### Author Rebuttal · Authors · 2025-03-31
>
> Thanks for reviewing the paper and checking the proofs.  We are quite disappointed by the gap between the assigned score (reject) and a review which sounds quite positive. We would appreciate further explanations on the concerns raised about lack of novelty: PW barycenters are new. However, we provide in the following detailed responses to the specific questions mentioned.
> > 1. Lack of comparisons with other methods
>
> The purpose of the archaeological section (S5) is not to propose a new morphological interpolation technique that outperforms the state of the art, but rather to demonstrate that in the OT univers, PW offers significant advantages compared with existing alternatives. In our archaeological application, the point clouds $X$ and $Y$ are approximately made of 10k points. At this scale, GW barycenters cannot be computed (well-known computational limitations). Nonetheless, we have conducted additional experiments by computing Wasserstein barycenters using the free support barycenters (FBS) method from the POT library, allowing for a comparison with the PW results shown in Figure 4. We will include these additional experiments in the appendix of the camera ready version of the paper. The computed Wasserstein barycenters can clearly be seen to be of lower quality, similarly to what is observed in Figure 1. Furthermore, we report below the corresponding objective functional values: Eq. (6) from our paper for PW barycenters, and Eq. (5) from [1] for the Wasserstein barycenters. The values for, respectively, the four interpolation steps are as follows:
> - Wasserstein (FBS): [0.00943, 0.01499, 0.01496, 0.00917]
> - PW: [0.00092, 0.00155, 0.00155, 0.00091]
>
> Our method achieves the lowest barycenter functional value for each interpolation step.
>
> > 2. In Figure 1, can the authors comment on the Gromov-Wasserstein results? Why would there be some outlier points?
>
> The GW barycenter is computed from pairwise distance matrices of size $N\times N$ and $M\times M$, where $N$ and $M$ correspond to the number of points in $X$ and $Y$, respectively. The barycenter itself is a $K\times K$ matrix (with $K$ fixed a priori) and must then be projected back into the original Euclidean space $\mathbb{R}^D$. In case of Figure 1, $D$ is set to 2. Dimensionality reduction techniques such as Multi-Dimensional Scaling (MDS, [3]) and t-Distributed Stochastic Neighbor Embedding (TSNE, [4]) can be used for this projection. However, this step can lead to suboptimal representations of the GW barycenter, even in simple cases such as the introductory example presented in Figure 1. We have made the caption of Figure 1 clearer in the final version of the paper (for more details please refer to the answer of Q1 for Reviewer HyoR).
>
>
> References:
>
> [1] Cuturi, Marco, and Arnaud Doucet. "Fast computation of Wasserstein barycenters." International conference on machine learning. PMLR, 2014.
>
> [2] Peyré, Gabriel, Marco Cuturi, and Justin Solomon. "Gromov-wasserstein averaging of kernel and distance matrices." International conference on machine learning. PMLR, 2016.
>
> [3] Borg, Ingwer, and Patrick JF Groenen. Modern multidimensional scaling: Theory and applications. Springer Science & Business Media, 2007.
>
> [4] Maaten, Laurens van der, and Geoffrey Hinton. "Visualizing data using t-SNE." Journal of machine learning research 9.Nov (2008): 2579-2605.
>
> [5] Cuturi, Marco. "Sinkhorn distances: Lightspeed computation of optimal transport." Advances in neural information processing systems 26 (2013).

---

> > ### Comment · Reviewer_nX3Q · 2025-04-05
> >
> > I appreciate the authors explaining the GW results in Figure 1, and acknowledge the contribution of propsing PW barycenter and algorithm. For comparisons in archaeological experiments, can the authors show the additional figure? Also, even when restricted to the OT universe, efficient alternatives include (but not limited to) the sliced Wasserstein barycenter [1,2], and the regularized Wasserstein Barycenter [3], deep Wasserstein embedding [4], sliced gromov Wasserstein [5], entropic gromov Wasserstein [6], and the low-rank arroximations [7]. I understand that it is not feasible to compare with all methods, but only one baseline is not enough.
> >
> > [1] Bonneel, Nicolas, et al. “Sliced and radon wasserstein barycenters of measures.” Journal of Mathematical Imaging and Vision 51.1 (2015): 22-45
> >
> > [2] Liutkus, A., Simsekli, U., Majewski, S., Durmus, A., & Stöter, F. R. (2019, May). Sliced-Wasserstein flows: Nonparametric generative modeling via optimal transport and diffusions. In International Conference on Machine Learning (pp. 4104-4113). PMLR.
> >
> > [3] Iterative Bregman projections for regularized transportation problems. SIAM Journal on Scientific Computing, 37(2), A1111-A1138.
> >
> > [4] Courty, Nicolas, Rémi Flamary, and Mélanie Ducoffe. "Learning wasserstein embeddings." arXiv preprint arXiv:1710.07457 (2017).
> >
> > [5] Titouan, Vayer, et al. "Sliced gromov-wasserstein." Advances in Neural Information Processing Systems 32 (2019).
> >
> > [6] Justin Solomon, Gabriel Peyr´ e, Vladimir G. Kim, and Suvrit Sra. Entropic metric alignment for correspondence problems. ACM Transactions on Graphics (TOG), 35(4):72:1–72:13, 2016.
> >
> > [7] Scetbon, Meyer, Gabriel Peyré, and Marco Cuturi. "Linear-time gromov wasserstein distances using low rank couplings and costs." International Conference on Machine Learning. PMLR, 2022.

---

> > > ### Author Response · Authors · 2025-04-08
> > >
> > > Thank you for your comments. Please find here the figure showing the Wasserstein barycenters computed for the archaeological section: https://pasteboard.co/LIjfirL63O7T.png
> > > > _“only one baseline is not enough”_
> > >
> > > We would like to emphasize that the main contributions of this paper lie in the new theoretical properties established for the PW distance, as well as in the novel barycenter formulation. As highlighted by Reviewer GNtQ, our experimental approach follows the OT barycenter literature. See for instance [8] with Wasserstein barycenters, and [9] introducing Gromov-Wasserstein barycenters, which focuses on qualitative evaluation through visual comparisons, without benchmarking. Nonetheless, we included Example 4.2 specifically to compare our barycenter with other state-of-the-art OT barycenters in a clustering scenario.
> > >
> > > > Barycenter comparisons
> > >
> > > Regarding the suggested references for the barycenter comparison, we would like to clarify that many of them are not directly applicable to the context of our application. Section 5 aims at finding barycentres and not at applying OT metrics for alignment, matching, etc. In order to present comparisons with other OT barycentres, we have computed (in the previous answer and shown in the above figure) the Wasserstein barycenters (Wb), which, however, do not represent a coherent bone structure. This follows naturally from the fact that the traditional Wasserstein problem is not invariant w.r.t. isometries in Euclidean space. Wb are thus unfit in cases of rotated and/or reflected distributions (bones in our application). [1] and [2] represent sliced variants (sWb) of the Wasserstein barycenter, which are again not invariant under isometries. sWb proved to be more computationally efficient than classical Wb, but often at the cost of geometric quality, particularly in local details. This effect can be seen also in Figure 6 of [1] with toy clouds. [3] involves entropic regularization via Sinkhorn iterations (regularized Wb), which indeed improves numerical stability but compromises the quality. In this regard, please also refer to our response to Question 2 of Review 6RWS for the formulation of Procrustes-Sinkhorn barycenters. [4] focuses on learning Wasserstein embeddings. The provided barycenters in [4] (e.g., mean of MNIST digits) are obtained via Euclidean averaging in a learned embedding space trained to approximate the Wasserstein distance. Furthermore, this embedding is not isometry-invariant. [5] introduces the sliced Gromov-Wasserstein distance (sGW). To the best of our knowledge sGW barycenters are not yet implemented in standard OT libraries such as POT, making comparisons nontrivial. However, it should be noticed that even in very simple scenarios such as the introductory example with birds, GW barycenters are subject to distortion. We clarified this in the previous answer. [6] focuses on non-rigid shape matching and [7] on the efficient calculation of GW distances. Both works do not define or study a notion of barycenter, which makes them less directly relevant to our analysis.
> > > To conclude and summarize, we respectfully point out that the list of references provided is almost entirely not relevant in terms of BARYCENTER comparison. We hope this clarifies.
> > >
> > > References:
> > >
> > > [1] Bonneel, Nicolas, et al. “Sliced and radon wasserstein barycenters of measures.” Journal of Mathematical Imaging and Vision 51.1 (2015): 22-45
> > >
> > > [2] Liutkus, A., Simsekli, U., Majewski, S., Durmus, A., & Stöter, F. R. (2019, May). Sliced-Wasserstein flows: Nonparametric generative modeling via optimal transport and diffusions. In International Conference on Machine Learning (pp. 4104-4113). PMLR.
> > >
> > > [3] Iterative Bregman projections for regularized transportation problems. SIAM Journal on Scientific Computing, 37(2), A1111-A1138.
> > >
> > > [4] Courty, Nicolas, Rémi Flamary, and Mélanie Ducoffe. "Learning wasserstein embeddings." arXiv preprint arXiv:1710.07457 (2017).
> > >
> > > [5] Titouan, Vayer, et al. "Sliced gromov-wasserstein." Advances in Neural Information Processing Systems 32 (2019).
> > >
> > > [6] Justin Solomon, Gabriel Peyr´ e, Vladimir G. Kim, and Suvrit Sra. Entropic metric alignment for correspondence problems. ACM Transactions on Graphics (TOG), 35(4):72:1–72:13, 2016.
> > >
> > > [7] Scetbon, Meyer, Gabriel Peyré, and Marco Cuturi. "Linear-time gromov wasserstein distances using low rank couplings and costs." International Conference on Machine Learning. PMLR, 2022.
> > >
> > > [8] Cuturi, Marco, and Arnaud Doucet. "Fast computation of Wasserstein barycenters." International conference on machine learning. PMLR, 2014.
> > >
> > > [9] Peyré, Gabriel, Marco Cuturi, and Justin Solomon. "Gromov-wasserstein averaging of kernel and distance matrices." International conference on machine learning. PMLR, 2016.

---

### Decision · Program_Chairs · 2025-05-01

**Decision:**

Accept (poster)

**Comment:**

The paper introduces the Procrustes-Wasserstein (PW) distance, a metric on the quotient space of discrete measures under rigid transformations. It extends Optimal Transport to account for pose alignment, proving PW is a valid distance and proposing algorithms for computing PW barycenters. The method is validated on synthetic data, MNIST clustering, and archaeological shape analysis, showing improved alignment and representative shape recovery. While most reviews are positive, one reviewer questioned the novelty and limited comparisons. In rebuttal, the authors emphasized that PW barycenters are new, clarified why many suggested baselines are not applicable due to lack of isometry invariance, and provided additional experimental results and visual comparisons.

While more extensive benchmarks could further strengthen the empirical side, the paper makes a meaningful theoretical and methodological contribution to the OT literature. I lean toward acceptance.